# Loss function of tumor suppressor FRMD8 confers resistance to tamoxifen therapy via a dual mechanism

Weijie Wu[1†], Miao Yu[1†], Qianchen Li[1], Yiqian Zhao[1], Lei Zhang[1], Yi Sun[1], Zhenbin Wang[1], Yuqing Gong[1], Wenjing Wang[2], Chenying Liu[3], Jing Zhang[1], Yan Tang[1], Xiaojie Xu[4]*, Xiaojing Guo[3]*, Jun Zhan[1]*, Hongquan Zhang[1]*

[1]Program for Cancer and Cell Biology, Department of Human Anatomy, Histology and Embryology, School of Basic Medical Sciences, Peking University International Cancer Institute, State Key Laboratory of Molecular Oncology, Peking University Health Science Center, Beijing, China; [2]Department of Gastroenterology, Beijing Friendship Hospital, Capital Medical University; State Key Laboratory for Digestive Health; National Clinical Research Center for Digestive Diseases, Beijing, China; [3]Department of Breast Pathology and Lab, Tianjin Medical University Cancer Institute and Hospital, National Clinical Research Center for Cancer, Key Laboratory of Breast Cancer Prevention and Therapy of Ministry of Education of China, Tianjin Medical University, Tianjin's Clinical Research Center for Cancer, Tianjin, China; [4]Department of Genetic Engineering, Beijing Institute of Biotechnology, Beijing, China

*For correspondence:
miraclexxj@126.com (XX);
guoxiaojing@tjmuch.com (XG);
zhanjun@bjmu.edu.cn (JZ);
hongquan.zhang@bjmu.edu.cn (HZ)

†These authors contributed equally to this work

**Competing interest:** The authors declare that no competing interests exist.

## eLife Assessment

The research presents **valuable** findings on the impact of FRMD8 loss on tumor progression and resistance to tamoxifen therapy. Through a series of **convincing** and systematic experiments, the author thoroughly investigates the role of FRMD8 in breast cancer and its underlying regulatory mechanisms. The study confirms that FRMD8 holds potential as a therapeutic target for reversing tamoxifen resistance, offering helpful insights for future treatment strategies.

**Abstract** Approximately 40% ERα-positive breast cancer patients suffer from therapeutic resistance to tamoxifen. Although reduced ERα level is the major cause of tamoxifen resistance, the underlying mechanisms remain elusive. Here, we report that FRMD8 raises the level of ERα at both transcriptional and post-translational layers. FRMD8 deficiency in *MMTV-Cre⁺*; *Frmd8ᶠˡ/ᶠˡ*; *PyMT* mice accelerates mammary tumor growth and loss of luminal phenotype, and confers tamoxifen resistance. Single-cell RNA profiling reveals that Frmd8 loss decreases the proportion of hormone-sensing differentiated epithelial cells and downregulates the levels of ERα. Mechanically, on one hand, loss of FRMD8 inhibits *ESR1* transcription via suppressing the expression of FOXO3A, a transcription factor of *ESR1*. On the other hand, FRMD8 interacts both with ERα and UBE3A, and disrupts the interaction of UBE3A with ERα, thereby blocking UBE3A-mediated ERα degradation. In breast cancer patients, *FRMD8* gene promoter is found hypermethylated and low level of FRMD8 predicts poor prognosis. Therefore, FRMD8 is an important regulator of ERα and may control therapeutic sensitivity to tamoxifen in ERα-positive breast cancer patients.

## Introduction

Breast cancer is the most commonly diagnosed cancer worldwide (*Sung et al., 2021*), and more than 70% of breast cancer are estrogen receptor α (ERα)-positive (*Habara and Shimada, 2022*). Although endocrine therapy is the most common systemic treatment for ERα-positive breast cancer in clinical practice, approximately 40% of patients still develop primary or secondary resistance to endocrine therapy (*Badia et al., 2007*; *Légaré and Basik, 2016*; *Rondón-Lagos et al., 2016*). Therefore, it is urgent and necessary to explore the mechanisms of endocrine therapy resistance and search for new therapeutic targets.

ERα is a ligand-activated transcription factor that is activated by estrogen and promotes cell proliferation during breast cancer development (*Harbeck et al., 2019*). Tamoxifen (TAM), a selective estrogen receptor antagonist, is the most widely used medicine in endocrine therapy. Tamoxifen competes with estrogen to bind with ERα and changes the conformation of ERα, thereby preventing the interaction between co-activators and ERα and inhibiting activation of ERα (*Katzenellenbogen et al., 2018*). Thus, the level of ERα is strongly correlated with reactivity and resistance of endocrine therapy. Uncovering the mechanisms by which ERα expression is regulated is essential for overcoming endocrine therapy resistance. Multiple transcription factors, such as AP-2γ, FOXO3, FOXM1, and GATA3, have been reported to bind to the promoter region of *ESR1*, the gene encoding ERα, and participate in transcriptional regulation of *ESR1* (*Jia et al., 2019*; *Kos et al., 2001*). In addition, post-translational modifications of ERα, including phosphorylation, acetylation, and ubiquitination, also have effects on subcellular localization, transcriptional activity, and stability of ERα (*Rogatsky et al., 1999*; *Williams et al., 2009*; *Zhou and Slingerland, 2014*). However, the mechanisms underlying the regulation of ERα expression are still not clear and require further investigation.

FERM domain-containing proteins are widely involved in processes such as the formation of macro-molecular complexes, subcellular localization, functional activation, and signal transduction, thereby regulating the occurrence and development of tumors (*Frame et al., 2010*; *Moleirinho et al., 2013*; *Zhan and Zhang, 2018*). FRMD8, as a member of FERM domain-containing proteins, has been reported to bind with iRhom and enhance the stability of the iRhom/TACE complex on the cell surface, thereby preventing iRhom/TACE degradation mediated by lysosomes. TACE is responsible for cleaving and releasing TNF, and the absence of FRMD8 impairs the production of TNF (*Künzel et al., 2018*; *Oikonomidi et al., 2018*). Additionally, FRMD8 expressions in both microenvironment and tumor cells promote lung tumor growth (*Badenes et al., 2023*). FRMD8 inhibits colon cancer growth by preventing cell cycle progression. FRMD8 disrupts the interaction of CDK7 with CDK4, subsequently inhibiting CDK4 activation. Furthermore, FRMD8 competes with MDM2 to bind RB, thereby attenuating MDM2-mediated RB degradation (*Yu et al., 2023*). However, the roles of FRMD8 in breast tumorigenesis and progression need further exploration.

In this study, we found that loss of Frmd8 in luminal epithelial cells of *MMTV-PyMT* mice accelerates mammary tumor progression and luminal epithelial phenotype loss, and confers tamoxifen resistance. Single-cell RNA profiling reveals that the number of hormone-sensing differentiated cells is diminished and the level of ERα is decreased in Frmd8-knocked-out mammary tumors. FRMD8 not only increases *ESR1* expression, but also prevents ERα degradation by interrupting the interaction between ERα and the E3 ligase UBE3A. Further, a low FRMD8 level predicts poor prognosis in human breast cancer patients. Thus, we demonstrated that FRMD8 is an important ERα regulator and a vital tumor suppressive protein in breast cancer growth and drug resistance.

## Results

### Loss of Frmd8 promotes mammary tumor growth and generates tamoxifen resistance *in vivo*

To examine whether FRMD8 plays a role in breast tumorigenesis, we established luminal epithelium-specific *Frmd8* knockout mice (*MMTV-Cre⁺*; *Frmd8^fl/fl^*) (*Figure 1A and B*) and further generated a Frmd8-deletion breast cancer mouse model (*MMTV-Cre⁺*; *Frmd8^fl/fl^*; *PyMT*) by crossing *MMTV-Cre⁺*; *Frmd8^fl/fl^* mice with *MMTV-PyMT* (*PyMT*) mice, a widely used transgenic mouse model of mammary tumorigenesis (*Figure 1C and D*, *Figure 1—figure supplement 1A and B*). Compared with *MMTV-Cre⁻*; *Frmd8^fl/fl^*; *PyMT* mice, Frmd8 depleted in mice significantly promotes mammary tumor development (*Figure 1E and F*). The total tumor weight (*Figure 1G*) and number of tumors (*Figure 1H*) were

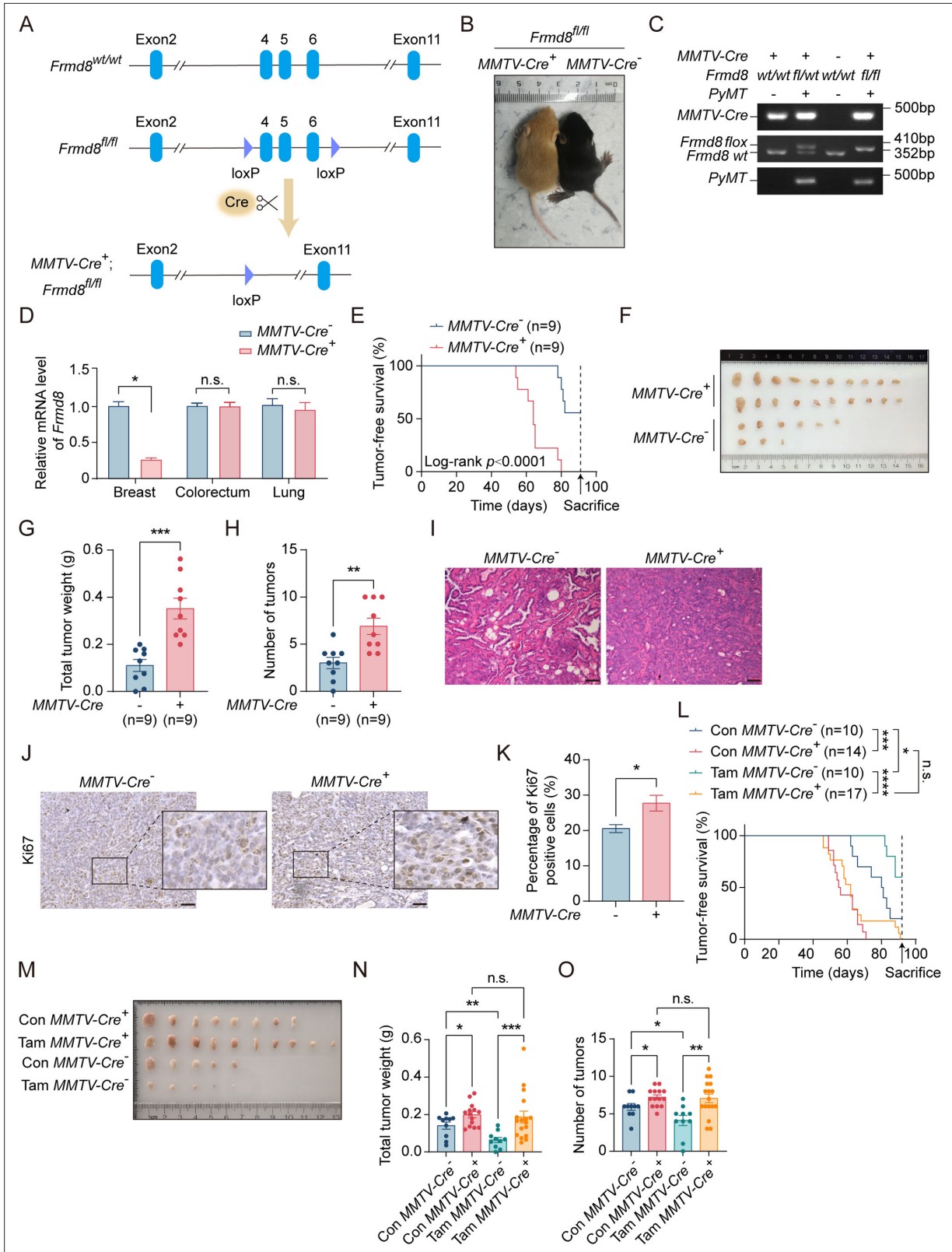

**Figure 1.** Loss of Frmd8 promotes mammary tumor growth and generates tamoxifen resistance *in vivo*. (**A**) A diagram of the *Frmd8* targeted alleles. Exons 4, 5, and 6 are flanked by loxp sites. (**B**) Distinguishing the genotype of littermate mice by mice coat color. Yellow represents *MMTV-Cre+; Frmd8fl/fl; PyMT* genotype, and black represents *MMTV-Cre-; Frmd8fl/fl; PyMT* genotype. (**C**) Representative PCR genotyping of mouse tail DNA. (**D**) Relative mRNA level of *Frmd8* in 7-week-old mammary glands from *PyMT* mice was analyzed by qRT-PCR. *Gapdh* was used as an internal reference.

*Figure 1 continued on next page*

*Figure 1 continued*

*p<0.05 by unpaired Student's *t*-test, n = 3. (**E**) Kaplan–Meier plot showing the appearance of palpable tumors in *MMTV-Cre⁻*; *Frmd8^{fl/fl}*; *PyMT* (n = 9) and *MMTV-Cre⁺*; *Frmd8^{fl/fl}*; *PyMT* (n = 9) mice (Log-rank test). (**F**) Representative images of tumors from *MMTV-Cre⁻*; *Frmd8^{fl/fl}*; *PyMT* and *MMTV-Cre⁺*; *Frmd8^{fl/fl}*; *PyMT* mice. (**G, H**) Total tumor weight (**G**) and number of tumors (**H**) per mice were measured. **p<0.01, ***p<0.001 by unpaired Student's *t*-test. (**I**) Representative H&E staining of tumors from *MMTV-Cre⁻*; *Frmd8^{fl/fl}*; *PyMT* and *MMTV-Cre⁺*; *Frmd8^{fl/fl}*; *PyMT* mice. Scale bar, 50 μm. (**J**) Immunohistochemistry (IHC) staining for Ki67 expression in mammary tumors from *PyMT* mice. The black boxes represent the magnified typical staining of the original images. Scale bar, 50 μm. (**K**) Quantification of Ki67-positive cell percentage in (**J**). *p<0.05 by unpaired Student's *t*-test, n = 3. (**L**) Kaplan–Meier plot showing the appearance of palpable tumors in *MMTV-Cre⁻*; *Frmd8^{fl/fl}*; *PyMT* and *MMTV-Cre⁺*; *Frmd8^{fl/fl}*; *PyMT* mice, with or without tamoxifen treatment (Log-rank test, *p<0.05, ***p<0.001, ****p<0.0001). (**M**) Representative images of tumors from *MMTV-Cre⁻*; *Frmd8^{fl/fl}*; *PyMT* and *MMTV-Cre⁺*; *Frmd8^{fl/fl}*; *PyMT* mice, with or without tamoxifen treatment. (**N, O**) Total tumor weight (**N**) and number of tumors (**O**) per mice from (**L**) were measured (n ≥ 10). *p<0.05, **p<0.01, ***p<0.001 by unpaired Student's *t*-test or Mann–Whitney test.

The online version of this article includes the following source data and figure supplement(s) for figure 1:

**Source data 1.** Unedited gel pictures for *Figure 1*, indicating the relevant bands.

**Source data 2.** Original files for gel pictures displayed in *Figure 1*.

**Figure supplement 1.** Loss of FRMD8 promotes breast cancer cell proliferation.

**Figure supplement 1—source data 1.** Unedited western blot pictures for *Figure 1—figure supplement 1*, indicating the relevant bands.

**Figure supplement 1—source data 2.** Original files for western blot pictures displayed in *Figure 1—figure supplement 1*.

markedly higher in *MMTV-Cre⁺*; *Frmd8^{fl/fl}*; *PyMT* mice than the control mice. Histological examination of the breast tumors from *PyMT* mice using H&E staining revealed that the luminal epithelium was poorly differentiated in *MMTV-Cre⁺*; *Frmd8^{fl/fl}*; *PyMT* mice tumors compared with the control mice (*Figure 1I*). Furthermore, immunohistochemical staining showed that the percentage of Ki67-positive cells was significantly elevated in mammary tumors of Frmd8-depleted mice (*Figure 1J and K*). These findings indicated that Frmd8 deficiency in the luminal epithelium accelerates mammary tumor growth in *MMTV-PyMT* mice and promotes cell proliferation.

Since tamoxifen is commonly used for the treatment of ERα⁺/HER2⁻ breast cancer, we thus investigated whether loss of Frmd8 affects sensitivity of mammary tumors to tamoxifen treatment in mice. To this end, *MMTV-Cre⁻*; *Frmd8^{fl/fl}*; *PyMT* mice and *MMTV-Cre⁺*; *Frmd8^{fl/fl}*; *PyMT* mice were injected intraperitoneally with tamoxifen or corn oil as control every 2 days. The results showed that tamoxifen significantly prevents mammary tumor progression in the control mice (*Figure 1L and M*). However, mammary tumors of *MMTV-Cre⁺*; *Frmd8^{fl/fl}*; *PyMT* mice showed no response to tamoxifen treatment (*Figure 1L and M*). Consistently, the total tumor weight (*Figure 1N*) and number of tumors (*Figure 1O*) of *MMTV-Cre⁻*; *Frmd8^{fl/fl}*; *PyMT* mice were markedly decreased after tamoxifen treatment, whereas there was no difference in *MMTV-Cre⁺*; *Frmd8^{fl/fl}*; *PyMT* mice (*Figure 1N and O*). Taken together, these findings demonstrated that loss of Frmd8 accelerates mammary tumor growth and generates resistance to tamoxifen therapy in Frmd8-depleted mice.

## Frmd8 knockout decreases the proportion of the hormone-sensing differentiated epithelial cells

To investigate the mechanism through which Frmd8 loss promotes mammary tumor growth and leads to tamoxifen resistance, we then performed single-cell RNA sequencing (scRNA-seq) analysis. Mammary tumors from 4-month-old *MMTV-Cre⁻*; *Frmd8^{fl/fl}*; *PyMT* and *MMTV-Cre⁺*; *Frmd8^{fl/fl}*; *PyMT* mice were harvested and analyzed using the Chromium Single Cell 3' Reagent Kitsv3 (10× Genomics). Cells that passed quality control (QC) filter totaled 24,320, of which 11,606 cells were from *MMTV-Cre⁻*; *Frmd8^{fl/fl}*; *PyMT*, and 12,714 cells were from *MMTV -Cre⁺*; *Frmd8^{fl/fl}*; *PyMT* mice (*Figure 2A*). After that, we profiled the three major cell lineages, including epithelial cells, immune cells, and stromal cells by the UMAP visualization (*Figure 2B*). Their associated top-expressed and canonical markers are shown in *Figure 2—figure supplement 1A and B*, respectively. Based on the expression of known markers, a total of 12 clearly separated cell lineages were finally identified (*Figure 2C and D*). In particular, they were as follows: B cells highly expressing *Cd19*, *Cd79a*, *Cd79b*; CD4⁺ T cells characterized with high *Cd3g* and *Cd4* expression; CD8⁺ T cells highly expressing *Cd3g* and *Cd8a*; dendritic cells (DCs) expressing *Cd74* and *Cd14*; endothelial cells specifically expressing the markers *Pecam1* and *Emcn*; epithelial cells expressing *Epcam* and *Krt8*; fibroblast cells high expressing *Col1a1* and *Col3a1*; granulocyte cells specifically expressing the markers *S100a9*; macrophage cells highly expressing *Cd14*,

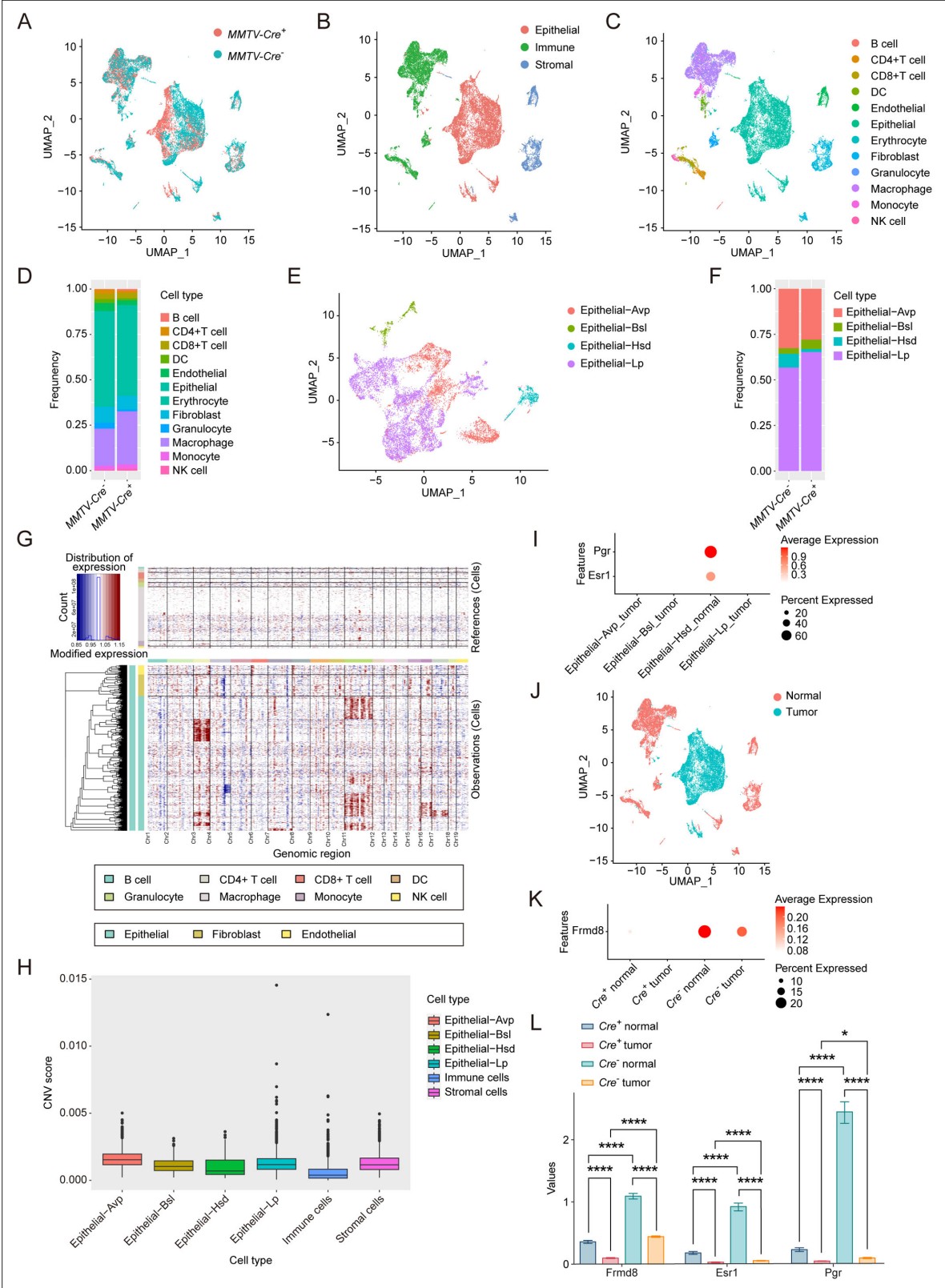

**Figure 2.** Frmd8 knockout decreases the proportion of the hormone-sensing differentiated epithelial cells. (**A**) T-SNE plot showing the distribution of cells from *MMTV-Cre⁻*; *Frmd8^fl/fl*; *PyMT* and *MMTV-Cre⁺*; *Frmd8^fl/fl*; *PyMT* mice. (**B**) T-SNE plot showing the distribution of epithelial, immune, and stromal cells. (**C, D**) T-SNE plot showing the distribution of main cell lineages (**C**) and their relative percentage (**D**). (**E, F**) T-SNE plot showing the distribution of epithelial cell lineages (**E**) and their relative percentage (**F**). (**G**) Heatmap showing distinct features of each cell lineages. Rows, genes. Columns, cells.

*Figure 2 continued on next page*

*Figure 2 continued*

The color key from blue to red indicates low to high gene expression. (**H**) Boxplot showing copy number variation (CNV) score of main cell lineages. (**I**) Dot plot showing the expression of *Esr1* and *Pgr* in epithelial cell lineages. (**J**) T-SNE plot showing the distribution of normal cells and tumor cells. (**K**) Dot plot showing the expression of *Frmd8* in normal and tumor cells from *MMTV-Cre⁻; Frmd8^fl/fl; PyMT* and *MMTV-Cre⁺; Frmd8^fl/fl; PyMT* mice. (**L**) Statistical analysis of *Frmd8, Esr1* and *Pgr* expression in normal and tumor cells from *MMTV-Cre⁻; Frmd8^fl/fl; PyMT* and *MMTV-Cre⁺; Frmd8^fl/fl; PyMT* mice. *p<0.05, ****p<0.0001 by Mann–Whitney test.

The online version of this article includes the following figure supplement(s) for figure 2:

**Figure supplement 1.** Distinct cell lineages determined by single-cell RNA-seq analysis.

*Cd68* and *C1qa*; monocyte cells highly expressing *Cd14* and *Ccr2*; and natural killer (NK) cells specifically expressing *Nkg7* and *Ncr1* (***Figure 2—figure supplement 1C***). The dot plots compared the proportion of cells expressing cluster-specific markers and their scaled relative expression levels (***Figure 2—figure supplement 1D***).

To further explore whether the proportions of epithelial cells in mammary tumors were affected by Frmd8 loss, we subset and reidentified four epithelial cell clusters (***Valdés-Mora et al., 2021***), including alveolar progenitor (Avp), basal (Bsl), hormone-sensing differentiated (Hsd), and luminal progenitor (Lp) epithelial cells (***Figure 2E***). As expected, the epithelial cells were composed of luminal epithelial cells and basal epithelial cells, which were consistent with the cellular characteristics of the mammary glands. In contrast to control mice, the proportion of the Hsd epithelial cells was significantly decreased in *MMTV-Cre⁺; Frmd8^fl/fl; PyMT* mice (***Figure 2F***).

To define the tumor cells in the mouse mammary tumors, we applied inferCNV algorithm to calculate the copy number variations (CNVs) of the single cells (***Figure 2G***). To this end, we analyzed CNV scores of epithelial subclusters, immune and stromal cells, revealing immune cells with low CNV score (***Figure 2H***). Given that Hsd epithelial cells that specifically expressed *Esr1* and *Pgr* had lower CNV score than alveolar progenitor (Avp), basal (Bsl), and luminal progenitor (Lp) epithelial cells, we defined the low CNV score Hsd epithelial cells as normal cells (***Figure 2H and I***). Furthermore, we defined the high CNV score epithelial cells as tumor cells and plotted the UMAP visualization (***Figure 2J***). The results also showed that the expression of *Frmd8* was decreased in tumor cells compared with normal cells (***Figure 2—figure supplement 1K and E***). In addition, we observed that loss of Frmd8 significantly decreased the expression of *Esr1* and *Pgr* in normal cells of mammary tumors and decreased expression of Frmd8 in tumor cells accompanied with low expression of *Esr1* and *Pgr* compared with normal cells (***Figure 2L***, ***Figure 2—figure supplement 1E***). Taken together, these findings indicated that Frmd8 depletion in *PyMT* mice leads to decreases of the Hsd epithelial cells proportion and the expression of *Esr1* and *Pgr*.

## FRMD8 promotion of *ESR1* expression is mediated by FOXO3A

Given that scRNA-seq results suggested that loss of Frmd8 reduced the proportion of Hsd epithelial cells and the expression of *Esr1* and *Pgr*, multiple immunofluorescence staining analyses were then performed to examine the change of ERα and PR at the protein levels *in situ* in mammary tumors of 4-month-old Frmd8-depleted mice. Consistent with the scRNA-seq results, deficiency of Frmd8 dramatically decreased the levels of ERα, PR, and CK8, a marker of mammary luminal epithelium, in normal tissues adjacent to tumors (***Figure 3A and B***). *In situ* tissue flow cytometry analysis demonstrated that in normal tissues adjacent to tumor of the control mice, the proportion of ERα-positive cells among CK8⁺ cells were 45.49%, whereas this proportion was just 9.28% in *MMTV-Cre⁺; Frmd8^fl/fl; PyMT* mice (***Figure 3—figure supplement 1A***). Similarly, the proportion of PR-positive cells among CK8⁺ cells was also reduced in normal tissues adjacent to tumor of *MMTV-Cre⁺; Frmd8^fl/fl; PyMT* mice compared with the control mice (***Figure 3—figure supplement 1B***). FRMD8^high ERα^high and FRMD8^high-PR^high cells were mainly present in normal tissues adjacent to tumor of *MMTV-Cre⁻; Frmd8^fl/fl; PyMT* mice, whereas Frmd8 depletion led to FRMD8^low ERα^low and FRMD8^low PR^low cells markedly increased (***Figure 3—figure supplement 1C and D***). In mammary tumor tissues from both control mice and *MMTV-Cre⁺; Frmd8^fl/fl; PyMT* mice, the expressions of ERα and PR were almost negative (***Figure 3A and B***). Since the expression of ERα is loss with tumor progression, to further clarify the regulation of Frmd8 on ERα, we performed immunohistochemistry (IHC) staining of mammary glands of 7-week-old *PyMT* mice, which had no palpable tumors. The results showed that Frmd8 depletion also markedly decreased the expression of ERα in normal and atypical hyperplasia breast tissues (***Figure 3—figure***

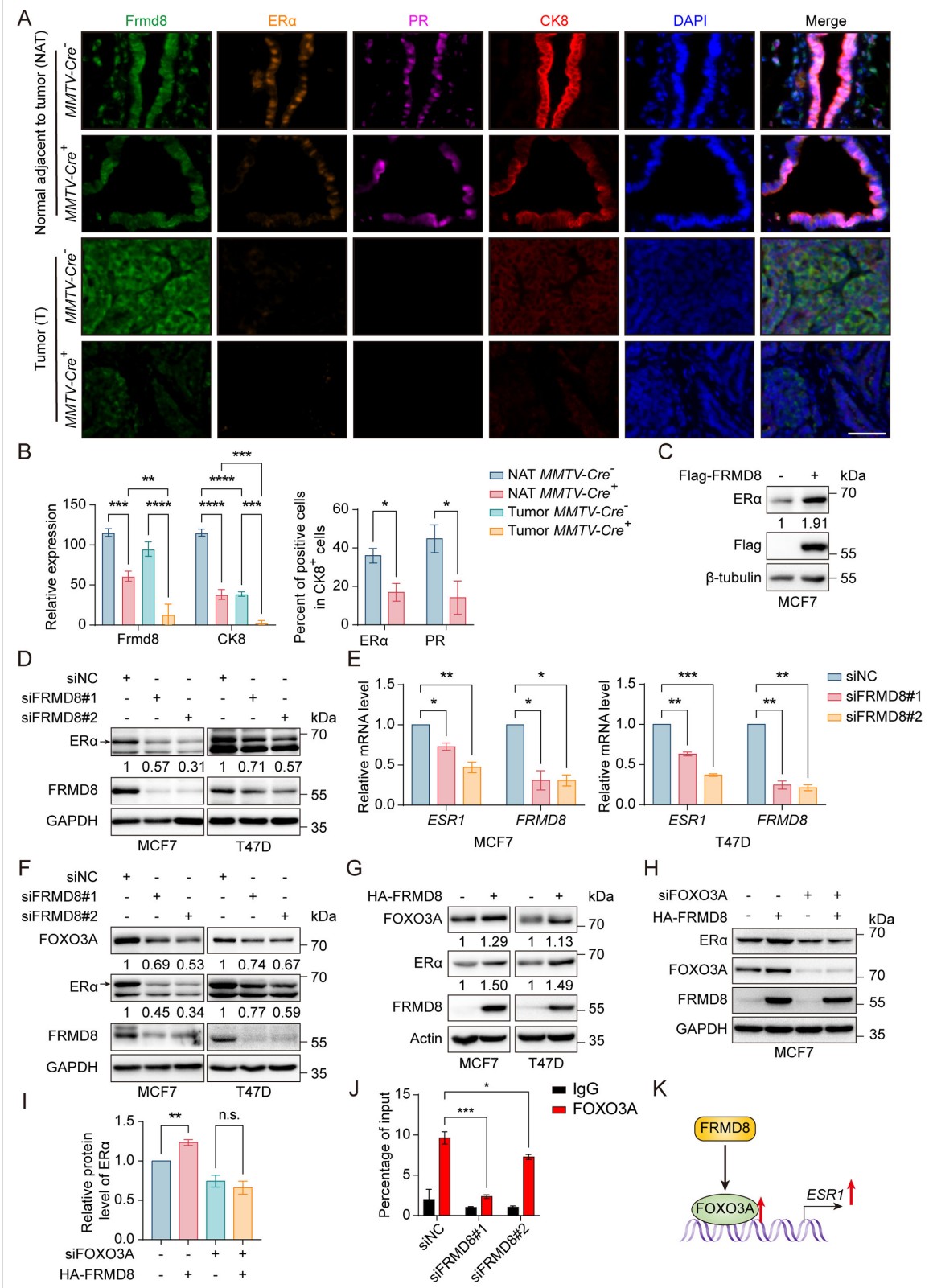

**Figure 3.** FRMD8 promotion of *ESR1* expression is mediated by FOXO3A. (**A**) Representative multiplex immunofluorescence images of tumor tissues and tissues adjacent to tumor from *MMTV-Cre⁻; Frmd8^fl/fl; PyMT* and *MMTV-Cre⁺; Frmd8^fl/fl; PyMT* mice. Scale bar, 50 μm. (**B**) Qualification of Frmd8 and CK8 expression (left panel) and ERα and PR-positive cell percentage in CK8⁺ cells (right panel) in (**A**). *p<0.05, **p<0.01, ***p<0.001, ****p<0.0001 by one-way ANOVA (left panel) or unpaired Student's *t*-test (right panel), n ≥ 3. (**C**) Lysates from MCF7 cells transiently transfected with Flag or Flag-

*Figure 3 continued on next page*

*Figure 3 continued*

FRMD8 were immunoblotted. (**D**) Lysates from MCF7 and T47D cells transiently transfected with control or FRMD8 siRNA were immunoblotted. In this and subsequent figures, specific bands are marked with an arrow. (**E**) Relative mRNA levels of *ESR1* and *FRMD8* from MCF7 and T47D cells transiently transfected with control or FRMD8 siRNA were analyzed by qRT-PCR. *GAPDH* was used as an internal reference. *p<0.05, **p<0.01, ***p<0.001 by one-way ANOVA, n = 2. (**F**) Lysates from MCF7 and T47D cells transiently transfected with control or FRMD8 siRNA were immunoblotted. (**G**) Lysates from MCF7 and T47D cells transiently transfected with HA or HA-FRMD8 were immunoblotted. (**H, I**) Lysates of MCF7 cells co-transfected with HA-FRMD8 and FOXO3A siRNA as indicated were immunoblotted (**H**). ERα protein levels were quantified by normalizing to the intensity of the GAPDH band (**I**). **p<0.01 by unpaired Student's *t*-test, n = 3. (**J**) Lysates of T47D cells transfected with control or FRMD8 siRNA were subjected to anti-FOXO3A ChIP-qPCR. *p<0.05, ***p<0.001 by one-way ANOVA, n = 2. (**K**) Working model for FRMD8 promotes *ESR1* transcription via upregulating FOXO3A expression.

The online version of this article includes the following source data and figure supplement(s) for figure 3:

**Source data 1.** Unedited western blot pictures for *Figure 3*, indicating the relevant bands.

**Source data 2.** Original files for western blot pictures displayed in *Figure 3*.

**Figure supplement 1.** The expressions of ERα and PR decreased in Frmd8-deleted mammary tissues.

*supplement 1E*). Taken together, these data suggested that the deficiency of FRMD8 downregulates the protein levels of ERα and PR in mammary tissues of *MMTV-PyMT* mice and accelerates the loss of luminal phenotype in mammary gland.

It was known that the common endocrine treatment for non-metastatic breast cancer relies on the expression of ERα (*Waks and Winer, 2019*), and our results demonstrated that loss of Frmd8 promotes mammary tumor growth and confers tamoxifen resistance in mice. We thus aim to further examine whether FRMD8 regulates ERα expression. To this end, we transiently transfect Flag-FRMD8 vector or FRMD8 siRNA in human breast cancer cells. Our results showed that FRMD8 overexpression drastically increased the levels of ERα, while ERα expression was greatly downregulated when FRMD8 was knocked down (*Figure 3C and D*). Moreover, qRT-PCR results indicated that depletion of FRMD8 significantly decreased the mRNA level of *ESR1* (*Figure 3E*).

FOXO3A is a crucial transcription factor for *ESR1* (*Jia et al., 2019*). To answer whether FOXO3A is involved in the regulation of FRMD8 on ERα, we examined the expression of FOXO3A after silencing FRMD8 through transfecting siRNA into MCF7 and T47D cells. The results showed that FRMD8 silencing dramatically decreased the level of FOXO3A (*Figure 3F*). Consistently, overexpression of FRMD8 in MCF7 and T47D cells markedly raised the level of FOXO3A (*Figure 3G*). To examine whether FRMD8 promotes *ESR1* transcription through FOXO3A, HA-tagged FRMD8 as well as FOXO3A siRNA were co-transfected into MCF7 cells. Although exogenous FRMD8 significantly upregulated ERα expression, depletion of endogenous FOXO3A greatly reduced the effect of FRMD8 on ERα (*Figure 3H and I*). Besides, chromatin immunoprecipitation (ChIP)-qPCR was performed and revealed that depletion of FRMD8 significantly decreased the occupancy of FOXO3A at the *ESR1* promoters (*Figure 3J*). Altogether, these data indicated that FRMD8 promoted upregulation of *ESR1* is mediated by FOXO3A (*Figure 3K*).

## FRMD8 stabilizes ERα via prevention of its degradation

Although our results suggest that FRMD8 depletion inhibits the mRNA level of *ESR1*, unexpectedly, we also observed that decreased expression of FRMD8 led to a decreased level of exogenous Flag-ERα (*Figure 4A*). This result suggested that in addition to inhibiting mRNA expression of ERα, FRMD8 may also regulate ERα protein expression at the post-translational level. To this end, T47D cells were treated with cycloheximide (CHX), a protein synthesis inhibitor. Depletion of FRMD8 greatly increased the ERα turnover rate and the half-life of ERα decreased from 6 h to approximately 2 h (*Figure 4B and C*). Furthermore, MCF7 and T47D cells were treated with MG132, a proteasome inhibitor, and chloroquine (CQ), a lysosome inhibitor, to determine whether ERα degradation was mediated by the proteasome or the lysosome. The results showed that the reduction of ERα levels by FRMD8 depletion was blocked by treatment with MG132 but not CQ (*Figure 4D*). Altogether, these data suggested that FRMD8 stabilizes ERα protein by inhibiting its degradation via a proteasome-mediated pathway.

## FRMD8 inhibits ERα degradation by blocking UBE3A binding with ERα

Given that FERM domain containing proteins play roles via regulating protein–protein interaction, we wonder whether there is an interaction between FRMD8 and ERα. To this end, co-immunoprecipitation

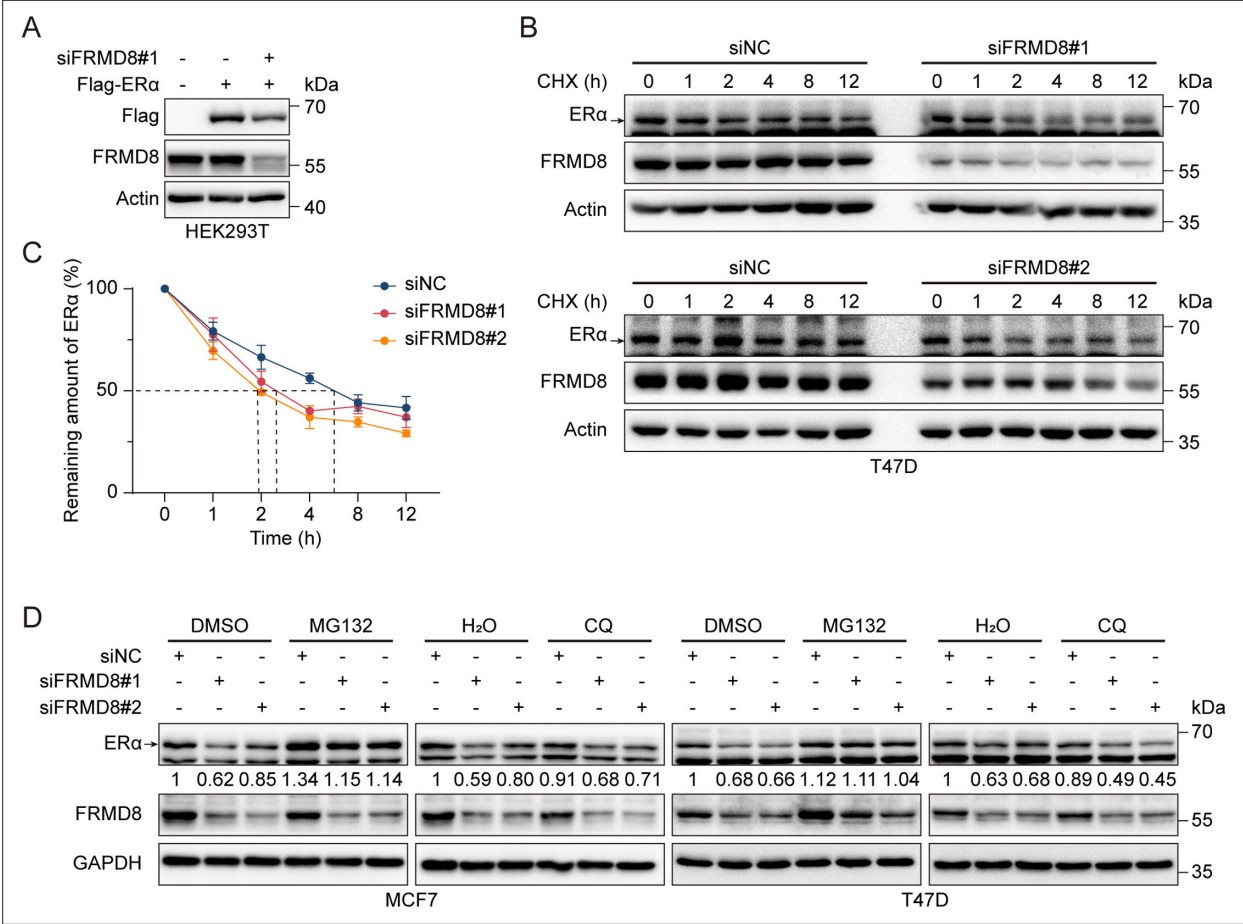

**Figure 4.** FRMD8 stabilizes ERα via prevention of its degradation. (**A**) Lysates of HEK293T cells co-transfected with Flag-ERα and FRMD8 siRNA as indicated were immunoblotted. (**B, C**) Lysates from T47D cells transiently transfected with control or FRMD8 siRNA were subjected to immunoblotting. Cells were treated with 100 μg/ml CHX for the indicated times (**B**). ERα protein levels were quantified by normalizing to the intensity of the Actin band, n = 4 (**C**). (**D**) Immunoblot analysis of ERα in MCF7 and T47D cells transiently transfected with control or FRMD8 siRNA. The cells were treated with MG132 (25 μM) or chloroquine (50 μM) for 6 h.

The online version of this article includes the following source data for figure 4:

**Source data 1.** Unedited western blot pictures for **Figure 4**, indicating the relevant bands.

**Source data 2.** Original files for western blot pictures displayed in **Figure 4**.

(co-IP) assays were performed using Flag-FRMD8 in MCF7 and T47D cells to examine the association of exogenous FRMD8 with endogenous ERα. The results showed that FRMD8 interacts with ERα (**Figure 5A and B**). Importantly, an interaction between endogenous FRMD8 and endogenous ERα was also observed in MCF7 cells (**Figure 5C**). Furthermore, a glutathione S-transferase (GST) pull-down assay using purified recombinant GST-FRMD8 and Flag-tagged ERα proteins was also performed. The results indicated that FRMD8 interacts directly with ERα *in vitro* (**Figure 5D**). Taken together, these findings suggest that FRMD8 is a binding partner of ERα in human cells.

The aforementioned findings demonstrated that FRMD8 prevents ERα degradation via proteasome pathway. To explore the mechanism by which FRMD8 inhibits ERα degradation, co-IP assay and mass spectrometry (MS) analysis were performed in HEK293A cells transiently expressing Flag-FRMD8 (**Supplementary file 1**). We next searched for the FRMD8-interacting proteins identified by MS matched with the known E3 ubiquitin ligases of ERα. Interestingly, UBE3A, a ubiquitin ligase for ERα (**Sun et al., 2012**), is the only matched protein that interacts with FRMD8 (**Figure 5E**). We thus examined the interaction between FRMD8 and UBE3A through co-IP. The results indicated an interaction between FRMD8 and UBE3A (**Figure 5F and G**). Further, we assumed that FRMD8 may interfere with the interaction between ERα and UBE3A. To test this idea, co-IP assays were performed in HEK293T

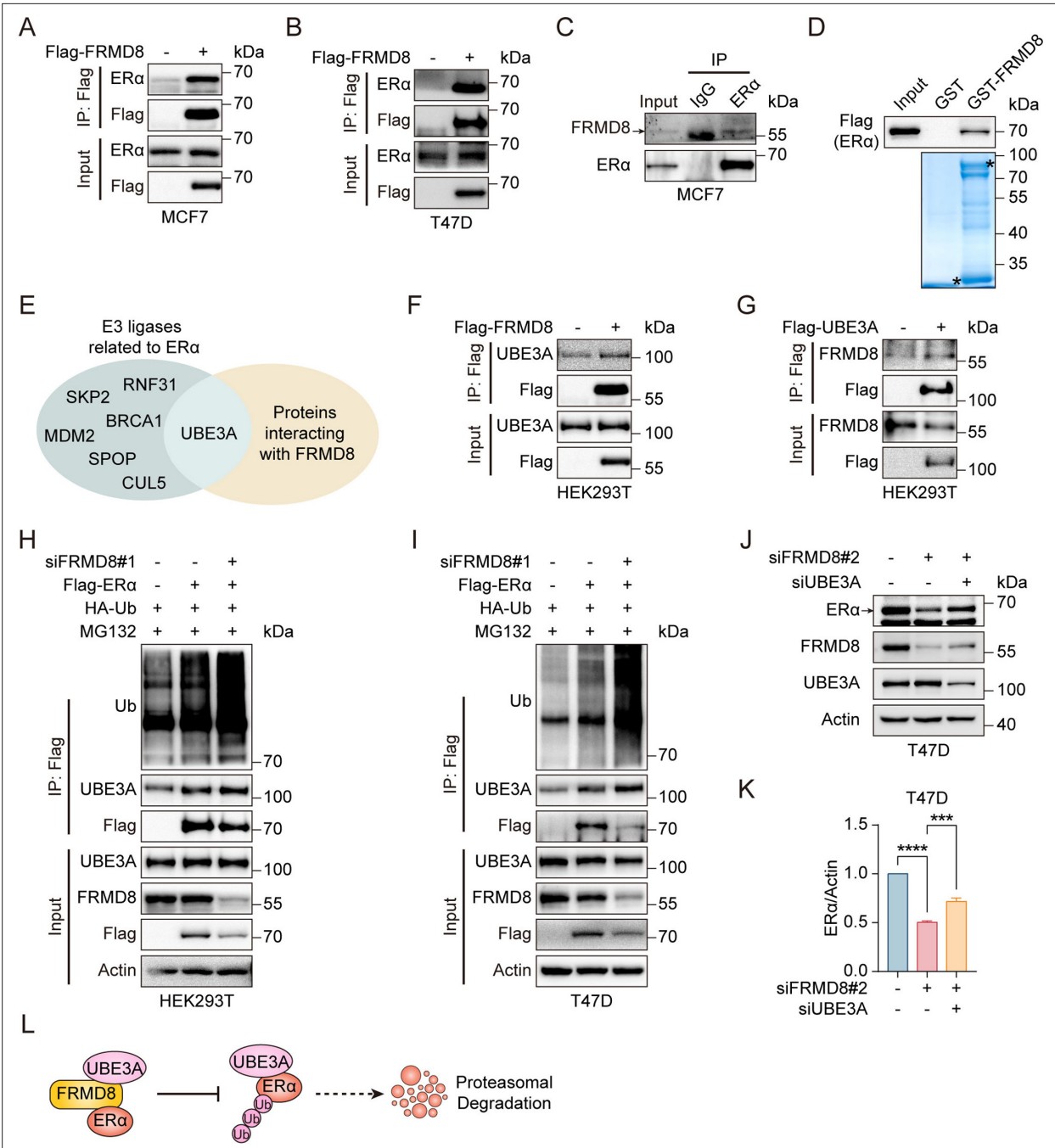

**Figure 5.** FRMD8 inhibits ERα degradation by blocking UBE3A binding with ERα. (**A, B**) Lysates of MCF7 (**A**) and T47D (**B**) cells transfected with Flag or Flag-FRMD8 were anti-Flag immunoprecipitated and immunoblotted for ERα and Flag. (**C**) Lysates from MCF7 cells were immunoprecipitated with IgG or anti-ERα, then immunoblotted for FRMD8 and ERα. (**D**) HEK293T cells were transiently transfected with Flag-ERα. ERα proteins in HEK293T whole-cell lysates (WCL) pulled down by GST or GST-FRMD8 recombinant proteins were subjected to western blot. Asterisks indicate proteins at the expected molecular weight. (**E**) Venn diagram showing overlap of E3 ligases related to ERα and proteins interacting with FRMD8. (**F**) Lysates of HEK293T cells transfected with Flag or Flag-FRMD8 were anti-Flag immunoprecipitated and immunoblotted. (**G**) Lysates of HEK293T cells transfected with Flag or Flag-UBE3A were anti-Flag immunoprecipitated and immunoblotted. (**H, I**) HEK293T (**H**) and T47D (**I**) cells were co-transfected with Flag-ERα, HA-Ub, and FRMD8 siRNA as indicated. Cells were treated with MG132 (25 μM) for 6 h. WCL were immunoprecipitated with anti-Flag and then immunoblotted for ubiquitinated ERα. (**J, K**) Lysates of T47D cells co-transfected with FRMD8 and UBE3A siRNA as indicated were immunoblotted (**J**). ERα protein levels were quantified by normalizing to the intensity of the Actin band (**K**). ***p<0.001, ****p<0.0001 by one-way ANOVA, n = 3. (**L**) Working model for FRMD8 disrupts the interaction between ERα and UBE3A, and protects ERα from UBE3A-mediated degradation.

The online version of this article includes the following source data for figure 5:

*Figure 5 continued on next page*

*Figure 5 continued*

**Source data 1.** Unedited western blot pictures for **Figure 5**, indicating the relevant bands.

**Source data 2.** Original files for western blot pictures displayed in **Figure 5**.

and T47D cells. In both cell types, the level of ERα ubiquitination and the interaction between ERα and UBE3A were markedly increased by FRMD8 depletion (**Figure 5H and I**). Intriguingly, FRMD8 depletion led to a marked decrease of ERα expression, while depletion of UBE3A dramatically rescued the effect of FRMD8 depletion (**Figure 5J and K**). Taken together, these results strongly demonstrated that FRMD8 binds to ERα and UBE3A, and prevents UBE3A interaction with ERα, thereby blocking UBE3A-mediated ERα ubiquitination and degradation (**Figure 5L**).

## *FRMD8* promoter is methylated and low FRMD8 level predicts poor prognosis in breast cancer patients

Given that Frmd8 prevents mammary tumor growth and inhibits tumor cell proliferation in mice (**Figure 1**), we would like to investigate whether the expression of FRMD8 is downregulated in mammary tumors. ScRNA-seq results demonstrated that the level of *Frmd8* in normal Hsd epithelial cells was significantly higher than other tumor cells in the mammary tumors from *PyMT* mice (**Figure 6A**). To further verify the expression of FRMD8 in human breast cancer cell lines, whole-cell lysates from normal mammary epithelial cells and various subtypes of breast cancer cell lines were subjected to western blot analysis. Results showed that FRMD8 expression was lower in claudin-low breast cancer cell lines compared with normal mammary epithelial cell line or ERα-positive cell lines (**Figure 6B**).

Hypermethylation of tumor suppressor genes is one of the major causes for tumorigenesis. Therefore, we investigated whether the promoter region of FRMD8 is hypermethylated in breast cancer patients. An analysis based on the TCGA database showed that the *FRMD8* promoter was highly methylated in primary breast tumors (**Figure 6C**). Subsequently, breast cancer cells were treated with the DNA methyltransferase inhibitor 5-Aza-2-deoxycytidine (5-Aza-dC). The results showed that 5-Aza-dC treatment significantly upregulated both mRNA and protein levels of FRMD8 in claudin-low but not luminal breast cancer cell lines (**Figure 6D and E**, **Figure 6—figure supplement 1A**), which was consistent with the lower expression of FRMD8 in claudin-low breast cancer cell lines (**Figure 6A**). These findings demonstrated that *FRMD8* gene promoter is hypermethylated, which could be the reason for the reduced FRMD8 expression in triple-negative breast cancer cells.

To further investigate the clinical significance of FRMD8 in breast cancer patients, we performed immunohistochemical staining of FRMD8 in a tissue microarray of breast cancer patients and evaluated the level of FRMD8 (**Figure 6F**). The results suggested that patients with lower level of FRMD8 showed poor overall survival (p=0.0409) (**Figure 6G**). Moreover, we also found that a decreased FRMD8 level was associated with poor recurrence-free survival in breast cancer patients according to Kaplan–Meier Plotter analysis (**Figure 6H and J**, **Figure 6—figure supplement 1B, C**). Collectively, these data indicated that the promoter of *FRMD8* is hypermethylated, and the low FRMD8 level predicts poor prognosis of breast cancer patients.

## Discussion

In this study, we found that FRMD8 plays a tumor suppressive role in breast cancer progression. We demonstrated that loss of FRMD8 promotes mammary tumor growth, accelerates the loss of mammary luminal phenotype, and confers tamoxifen resistance via downregulating ERα level. FRMD8 depletion not only suppresses transcription of *ESR1* through decreasing the level of FOXO3A, but also promotes the E3 ligase UBE3A binding with ERα to disrupts ERα protein stability. Moreover, the promoter of *FRMD8* is hypermethylated and a decreased FRMD8 level predicts poor outcomes in breast cancer patients (**Figure 7**).

As a FERM-domain containing protein, FRMD8 is necessary for releasing of TNF and EGFR ligands (**Künzel et al., 2018**; **Oikonomidi et al., 2018**). FRMD8 regulates lung cancer cell growth by regulating tumor microenvironment (**Badenes et al., 2023**). Frmd8-deficient mice are defective in intestinal epithelial barrier repair function and deletion of Frmd8 promotes colorectal tumorigenesis induced by

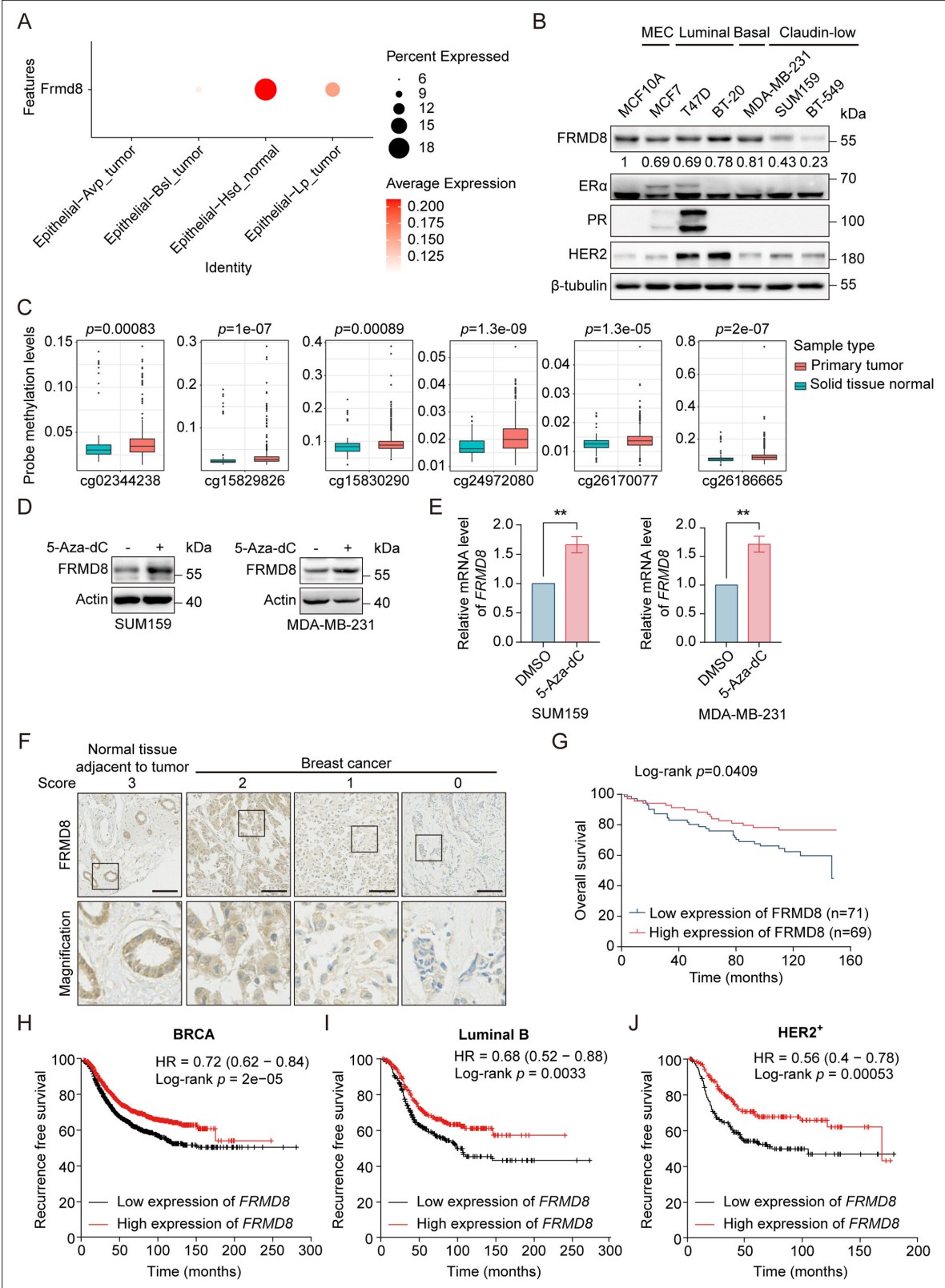

**Figure 6.** *FRMD8* promoter is methylated and low FRMD8 level predicts poor prognosis in breast cancer patients. (**A**) Dot plot showing the expression of *Frmd8* in epithelial cell lineages from *PyMT* mice. (**B**) Lysates from human mammary epithelial cell (MEC) and breast cancer cells were subjected to immunoblotting. (**C**) Methylation of *FRMD8* promoter region in breast cancer according to the University of California Santa Cruz (UCSC) database (http://xena.ucsc.edu/). (**D**) SUM159 and MDA-MB-231 cells were treated with 5-Aza-dC (10 µM) for 48 h. Protein expression of FRMD8 was examined

*Figure 6 continued on next page*

*Figure 6 continued*

by western blot. (**E**) SUM159 and MDA-MB-231 cells were treated with 5-Aza-dC (10 µM) for 48 h. *FRMD8* mRNA levels was examined by quantitative reverse transcription PCR (qRT-PCR) *GAPDH* was used as an internal reference. **p<0.01 by unpaired Student's *t*-test, n = 3. (**F**) IHC analysis of FRMD8 expression in human breast carcinoma TMA was performed. Representative examples (scale bar, 100 µm) of normal tissue adjacent to tumor and breast cancer with different levels of FRMD8 expression are shown, with the magnification of selected areas inserted. (**G**) Kaplan–Meier analysis for the overall survival of breast cancer patients according to FRMD8 expression (Log-rank test). (**H–J**) Recurrence-free survival of breast cancer patients according to *FRMD8* expression were analyzed according to Kaplan–Meier plotter (http://kmplot.com/analysis/).

The online version of this article includes the following source data and figure supplement(s) for figure 6:

**Source data 1.** Unedited western blot pictures for *Figure 6*, indicating the relevant bands.

**Source data 2.** Original files for western blot pictures displayed in *Figure 6*.

**Figure supplement 1.** *FRMD8* promoter is methylated and low FRMD8 level predicts poor prognosis in breast cancer patients.

**Figure supplement 1—source data 1.** Unedited western blot pictures for *Figure 6—figure supplement 1*, indicating the relevant bands.

**Figure supplement 1—source data 2.** Original files for western blot pictures displayed in *Figure 6—figure supplement 1*.

azoxymethane/ dextran sodium sulfate in mice (*Badenes et al., 2023*; *Yu et al., 2023*). FRMD8 acts as a scaffold protein, inhibiting CDK4 activation mediated by CDK7 and preventing MDM2-mediated RB degradation (*Yu et al., 2023*). In this study, our results suggested that FRMD8 inhibits mammary tumor progression in *MMTV-PyMT* mice. Besides, FRMD8 also acts as a scaffold molecule, which interacts simultaneously with ERα and UBE3A, and renders UBE3A unable to bind with ERα, thus stabilizing ERα.

In *MMTV-PyMT* mice, early-stage mammary tumors express ERα and PR, but these receptors are gradually lost as the tumor progresses (*Lapidus et al., 1998*). Our scRNA-seq results revealed that mammary tumor epithelial cells in *MMTV-PyMT* mice fall into four clusters, with only Hsd epithelial cells showing ERα and PR expression. Additionally, Hsd epithelial cells exhibited the lowest CNV score, indicating a closer resemblance to normal epithelial cells. The loss of Frmd8 reduced the proportion of Hsd epithelial cells and led to a downregulation of ERα and PR expression, implying that Frmd8 deficiency promotes the loss of luminal features in the mammary gland and accelerates mammary tumor progression.

In this study, we identified that loss of FRMD8 inhibiting *ESR1* expression through downregulating the level of FOXO3A. FOXO3A locates in nucleus and blocks cell cycle progression via activating the cell cycle blocking protein p27$^{KIP1}$ (*Seoane et al., 2004*). In addition, FOXO3A contributes to cell death through a Fas ligand-dependent mechanism (*Seoane et al., 2004*). Phosphorylated FOXO3A mediated by AKT binds to 14-3-3 protein and remains in the cytoplasm, resulting in loss of activity (*Arden, 2004*). Thus, FRMD8 may also be involved in the regulation of cell cycle or apoptosis in an ERα-independent manner by regulating FOXO3A. Furthermore, FOXO3A is closely associated with the stemness of breast cancer cells. FOXO3A suppresses breast cancer stem cell properties and tumorigenicity via inhibition of FOXM1/SOX2 signaling (*Liu et al., 2020*; *Yan et al., 2017*), suggesting that FRMD8 may have an effect on breast cancer stem cells. However, the mechanisms by which FRMD8 promotes FOXO3A expression remain unclear and need to be further investigated.

The proportion of luminal subtype is increasing in new cases of breast cancer (*Waks and Winer, 2019*). Tamoxifen therapy is one of the most important systemic treatment for ERα$^+$/HER2$^-$ subtype breast cancer patients (*Waks and Winer, 2019*). Unfortunately, primary or secondary tamoxifen resistance occurs in about 40% of patients treated with tamoxifen therapy, and tamoxifen resistance often leads to the development of resistance to other selective estrogen receptor modulators (*Badia et al., 2007*; *Légaré and Basik, 2016*; *Rondón-Lagos et al., 2016*). As a regulator of ERα, FRMD8 may be a therapeutic target for rescue tamoxifen resistance.

In summary, we identified FRMD8 as a prognostic marker in breast cancer. FRMD8 regulates ERα level through a dual mechanism. Loss of FRMD8 inhibits *ESR1* transcription via downregulating FOXO3A expression. FRMD8 also stabilizes ERα protein by preventing UBE3A from binding to ERα. Deficiency of FRMD8 promotes mammary luminal features loss and confers tamoxifen resistance. Our findings indicated that targeting *FRMD8* promoter methylation may provide a novel therapeutic approach for reversing tamoxifen resistance.

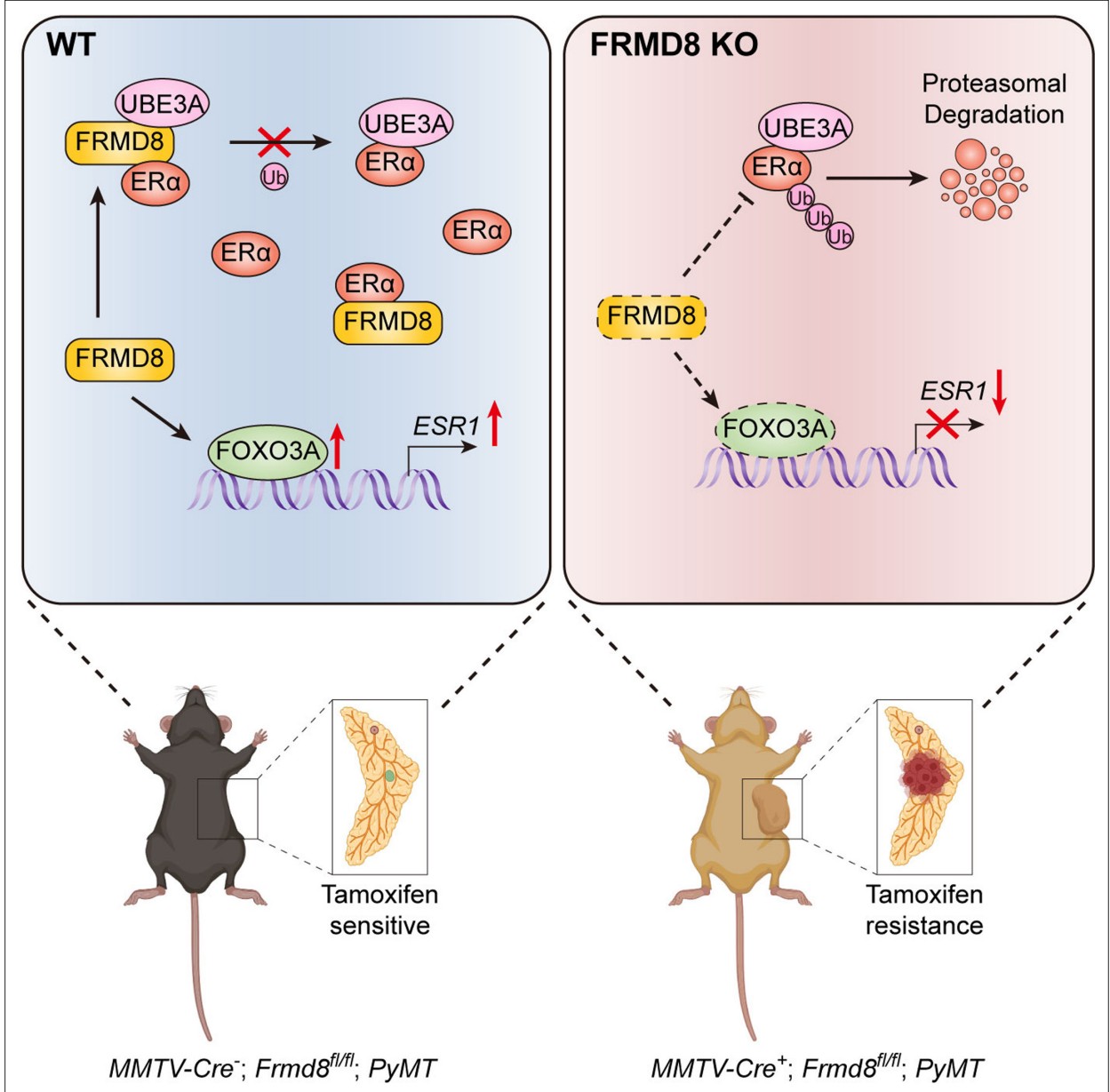

**Figure 7.** A working model shows that loss of FRMD8 promotes mammary tumor progression and confers tamoxifen resistance by downregulating ERα expression at both transcriptional and post-translational levels.

## Materials and methods
### Mice

*Frmd8* floxed mice and *MMTV-PyMT* mice, both on a C57BL/6 background, were purchased from the Shanghai Model Organisms Center Inc *MMTV-Cre+* were from the Nanjing Biomedical Research Institute of Nanjing University.

*Frmd8* floxed (*Frmd8fl/fl*) mice were crossed with *MMTV-Cre+*; *PyMT* transgenic mice to generate *Frmd8* heterozygous (*MMTV-Cre+*; *Frmd8fl/wt*; *PyMT*) mice. *MMTV-Cre+*; *Frmd8fl/wt*; *PyMT* mice were further backcrossed with *Frmd8fl/fl* mice to obtain littermate *MMTV-Cre-*; *Frmd8fl/fl*; *PyMT* and *MMTV-Cre+*; *Frmd8fl/fl*; *PyMT* mice.

Mice were housed in a pathogen-free animal facility at Laboratory Animal Center of Peking University Health Science Center with a 12 h light/dark cycle, constant temperature and humidity, and fed standard rodent chow and water *ad libitum*. All animal experiments were approved by the Peking

University Biomedical Ethics Committee and the approval number is BCJB0104. Genomic DNA extracted from mouse tail biopsies was subjected to standard genotyping PCR using the primers specified in *Supplementary file 2*. The reaction conditions were 5 min at 94°C; 35 cycles of 30 s at 94°C, 30 s at 56°C and 1 min at 72°C; followed by 5 min at 72°C and hold at 4°C.

## Tamoxifen treatment

Seven-week-old female mice were given intraperitoneal injections of either corn oil or tamoxifen (50 mg/kg) every 2 days. Tumor formation was assessed every 2 days, and the time point at which the tumor first became palpable was recorded as the tumor-free survival time. At the end of the treatment, mice were euthanized and the total number of tumors were counted. Tumors were measured using calipers, and tumor volume was calculated using the formula: $V$ = (length × width × height × 0.5) mm$^3$. Tumors were subsequently fixed in 4% paraformaldehyde (PFA) and housed in individual cassettes for paraffin embedding. 5 μm sections were stained for H&E or immunohistochemistry.

## Antibodies and reagents

Antibodies against ERα (#8644), Ubiquitin (#3936), and PR (#8757) were purchased from Cell Signaling Technology. Antibodies specific for ERα (#ab32063), CK8 (#ab53280), FOXO3A (#ab109629), UBE3A (#ab272168), and HER2 (#ab134182) were from Abcam. Anti-PR (#YM3348) antibody was from Immunoway. Anti-FRMD8 (#HPA002861) antibody was from Atlas Antibodies. Anti-Flag (#F3165) antibody was from Sigma-Aldrich. Antibodies against GAPDH (#AC002) were purchased from ABclonal Technology. Anti-Actin (#sc-58673) antibodies were from Santa Cruz Biotechnology. Antibodies specific for β-tubulin (#TA-10) were from Zhong Shan Jin Qiao (ZSGB-Bio).

MG132 (#S2619) was purchased from Selleck. CQ (#C6628), CHX (#C7698), and 5-Aza-dC (#A3656) were purchased from Sigma-Aldrich. Tamoxifen (#HY-13757A) and estradiol (E2, #HY-B0141) were purchased from MedChemExpress.

## Plasmids

The recombinant vectors encoding human FRMD8 and ERα were constructed by PCR-based amplification, and then subcloned into the p3×Flag-CMV-10 expression vector. The GST-tagged FRMD8 expression plasmid was generated by inserting PCR-amplified fragments into pGEX-4T-1 vector. All constructs were confirmed by DNA sequencing.

## Cell culture

HEK293T, HEK293A, BT-20, MDA-MB-231, SUM159, BT-549, and MCF10A were purchased from American Type Culture Collection. MCF7 and T47D were purchased from the Cell Resource Center, Peking Union Medical College (the headquarter of the National Infrastructure of Cell Line Resource). Cell lines were checked free of mycoplasma contamination by PCR and culture. Their species origin was confirmed with PCR. The identity of human cell lines was authenticated with STR profiling. Human embryonic kidney cell lines HEK293T and HEK293A and human breast cancer cell lines MCF7, BT-20, MDA-MB-231, and SUM159 were cultured in Dulbecco's Modified Eagle Medium (DMEM). Human breast cancer cell lines T47D and BT-549 were cultured in RPMI 1640 medium. All media were supplemented with 10% fetal bovine serum, 100 U/ml penicillin, and 0.1 mg/ml streptomycin. Medium for culturing T47D was supplemented with 0.2 U/ml insulin. Human breast cancer cell line MCF10A was cultured in DMEM/F12 medium supplemented with 10% horse serum, 100 U/ml penicillin and 0.1 mg/ml streptomycin, 20 ng/ml epidermal growth factor, 0.5 μg/ml hydrocortisone, 10 μg/ml insulin, and 100 ng/ml cholera toxin. All cell lines were maintained in a humidified atmosphere at 37°C with 5% $CO_2$ and passaged using 0.25% trypsin containing 0.02% EDTA for dissociation at 80% confluence.

## Plasmid and specific siRNA transfection

For transient transfection, cells at 50–60% confluence were transfected with plasmids via polyethylenimine (PEI; Polyscience, #24765). For RNA interference experiments, cells at 60–70% confluence were transfected with siRNA using Lipo8000 (Beyotime, #C0533). The specific sequences of siRNA are listed in *Supplementary file 2*.

## Extraction of proteins and immunoblotting

Cells or tissues were lysed using RIPA buffer (1×PBS, pH 7.4, 0.5% sodium deoxycholate, 1% NP-40, and 0.1% SDS) complemented with an EDTA-free cocktail of protease inhibitors (Roche). This was followed by centrifugation at 12,000 rpm for over 15 min at a temperature of 4°C to collect the supernatant. Protein concentrations were ascertained using a bicinchoninic acid (BCA) protein assay kit (Applygen, #P1511). Protein samples underwent electrophoresis via SDS-PAGE and were then transferred to a polyvinylidene fluoride membrane employing conventional methods. The membranes were treated with 5% non-fat milk for 1 h, then incubated with primary antibodies at 4°C overnight, and subsequently exposed to HRP-conjugated goat anti-mouse or rabbit IgG secondary antibodies for 1 h at 4°C. The bound antibodies were visualized using the EZ-ECL Chemiluminescence Detection Kit for HRP (Biological Industries, # 20-500-1000) through ChampChemi (SageCreation).

## Chromatin immunoprecipitation

The chromatin immunoprecipitation (ChIP) assay was performed according to the manufacturer's instructions of Sonication ChIP Kit (ABclonal, #RK20258). The purified DNA was analyzed by quantitative reverse transcription PCR (qRT-PCR). All primers are shown in *Supplementary file 2*.

## Immunoprecipitation

IgG, serving as controls, or indicated antibodies were added to pre-cleared lysates and allowed to incubate overnight at 4°C with sustained rocking. Following this, lysates were subjected to a 4 h incubation with 35–50 μl of either protein A or protein G Sepharose beads (Santa Cruz Biotechnology) at 4°C. Subsequently, the Sepharose beads were washed thrice with RIPA buffer and heated in SDS-loading buffer at 99°C for 10 min. The immunoprecipitation results were assessed via immunoblotting.

## Mass spectrometry analysis

Flag-FRMD8 and empty vector were transiently transfected in HEK293A cells using PEI, and cell lysates were harvested after 48 h. The cell lysates were pre-cleared and then incubated with pre-washed anti-Flag M2 agarose beads (Yeasen, #20584ES08) overnight at 4°C. Beads were washed three times with PBS. Protein samples were then eluted by boiling with sodium dodecyl sulfate (SDS)-loading buffer at 99°C for 10 min. 10% of the samples were saved for immunoblotting. The other 90% of samples were separated by SDS-PAGE, visualized by colloidal Coomassie blue staining, destained, and subjected to mass spectrometry (MS) analysis.

For MS analysis, the protein in the gel was subjected to trypsin digestion. In the LC-MS/MS analysis, the digested products were separated using a 120 min gradient elution at a flow rate of 0.300 μl/min, utilizing the Thermo Ultimate 3000 nano-UPLC system directly interfaced with the Thermo Fusion LUMOS mass spectrometer. The analytical column used was an Acclaim PepMap RSLC (75 μm ID, 250 mm length, C18). Mobile phase A consisted of 0.1% formic acid, while mobile phase B was 100% acetonitrile with 0.1% formic acid. The Fusion LUMOS mass spectrometer operated in data-dependent acquisition mode, employing Xcalibur 4.1.50 software. It started with a single full-scan mass spectrum in the Orbitrap (375–1500 m/z, 60,000 resolution), followed by data-dependent MS/MS scans. The MS/MS spectra from each LC-MS/MS run were analyzed against the selected database using Proteome Discovery software (version 2.4).

## Recombinant protein expression and GST pull-down assay

For the expression of glutathione S-transferase (GST) fusion proteins, GST-tagged recombinant vectors were transfected into *Escherichia coli* Rosetta (DE3) cells. Upon the cell culture attaining an $OD_{600}$ between 0.6 and 0.8, the cells were subjected to overnight incubation with 1 mM isopropyl-β-D-thiogalactopyranoside (Sigma, #92320) at 20°C. For the purification of GST fusion proteins, cells were collected and sonicated.

For GST-pull down assays, HEK293T cells were transfected and subsequently lysed on ice for over 15 min. Purified GST or GST fusion proteins were anchored onto Glutathione Sepharose 4B beads (Pharmacia Biotech). These beads were then further incubated with cellular extract at 4°C for an extended period of over 4 h, followed by washes with ice-cold PBS for three times. The process concluded with western blot analysis.

## RNA extraction and qRT-PCR

Total RNA was isolated using Trizol reagent (Invitrogen) as per the manufacturer's guidelines. 1 µg of the isolated RNA was converted to cDNA using HiScript II Q RT SuperMix for qPCR (+gDNA wiper) (Vazyme). Quantitative real-time PCR, carried out in triplicate, was used to assess relative mRNA levels and was standardized to *GAPDH* expression. The cDNA products were amplified using the LightCycler 96 (Roche) platform, and data were subsequently analyzed through LightCycler 96 (Roche) along with GraphPad Prism 7.0 software. All relevant primers are listed in *Supplementary file 2*.

## Histology and immunohistochemistry staining

Tissues were preserved in 4% PFA, embedded in paraffin, and then sliced into 5 µm sections prior to staining. Following deparaffinization and rehydration, these sections were subject to H&E staining for structural examination. Immunohistochemistry staining was executed using the streptavidin-biotin-peroxidase complex method, with subsequent detection of 3'3'-diaminobenzidine as guided by the manufacturer's instructions (Dako, Agilent Pathology Solutions). Observations and imaging were performed using an Olympus BX51 microscope coupled with an Olympus DP73 CCD photography system.

## Tumor tissue microarray (TMA) immunohistochemistry analysis

Human breast carcinoma TMA (HBre-Duc140Sur-01) was purchased from Shanghai Biochip. Patient information of TMA is listed in *Supplementary file 3*. Immunohistochemistry staining for FRMD8 in TMA was performed as described above. The stained microarrays were evaluated by a pathologist blind to cancer outcomes. Based on histological evaluations, staining reactivity was categorized into four levels: absence of reactivity (score = 0), weak reactivity (score = 1), moderate reactivity (score = 2), and intense reactivity (score = 3 or 4).

## Multiple immunohistochemistry staining and analysis

Multicolor immunohistochemistry was conducted using the TissueGnostics Multiple IHC Assay Kit (TissueGnostics, TGFP550) on mammary tumors from mice. Tumors were preserved in 4% PFA, embedded in paraffin, and then sliced into 5 µm sections prior to staining. The sections were then deparaffinized and hydrated, followed by antigen retrieval in a Tris-EDTA buffer via microwave. Slides underwent treatment with 3% $H_2O_2$ at ambient temperature to eliminate endogenous peroxidase activity. For iterative rounds of cyclic staining, slides were treated with blocking solution, then incubated with the primary antibody either overnight at 4°C or for 2 h at 37°C, and subsequently with HRP-linked secondary antibody at 37°C for 30 min. Signal was enhanced using Tyramide Signal Amplification (TSA) reagents. Antigen retrieval buffer was applied once more, and the aforementioned steps were repeated for additional staining. Lastly, DAPI was used for 10 min to stain nuclei. Tissue images were captured using the tissue faxs platform (TissueGnostics).

Immunofluorescence image quantification was carried out via StrataQuest v7.0.158 software (TissueGnostics). Cells were identified based on DAPI staining, and the expression levels of Frmd8, ERα, PR, and CK8 were calculated by the software, measuring fluorescence intensity and area to enumerate positively stained cells. The threshold for identifying positive cells was set by inspecting cell recognition on the original image using the View Backward Data function.

## Single-cell RNA sequence (scRNA-seq)

Single-cell RNA sequencing libraries were meticulously prepared utilizing the Chromium Single Cell 3' Reagent Kits v3 according to the protocols recommended by 10×Genomics. Briefly, an initial quantity of approximately $1 \times 10^5$ cells, which had been sorted through fluorescence-activated cell sorting, underwent a triple wash process in DPBS containing 0.04% BSA. These cells were subsequently resuspended to a final concentration of 700–1200 cells/µl, with a minimum viability threshold of 85%. During the library preparation process, cells were encapsulated within droplets to facilitate a precise targeted recovery rate. After the reverse transcription phase, the generated emulsions were disrupted, and the barcoded cDNA was isolated using Dynabeads technology. This was followed by PCR amplification to enrich the cDNA pool. The amplified cDNA served as the template for constructing the 3' gene expression libraries. Specifically, for this construction phase, 50 ng of the amplified cDNA was fragmented and subjected to end repair. This material was then subjected to a double-size selection

process using SPRIselect beads. The prepared libraries were sequenced on NovaSeq system provided by Illumina, generating 150 bp paired-end reads.

### Single-cell RNA sequence original data process

Prior to commencing the analysis, cellular data underwent a stringent filtering process. Cells with unique molecular identifier counts below 30,000 and those exhibiting gene counts within the range of 200–5000 were excluded. Additionally, cells displaying a mitochondrial content exceeding 20% were also removed from further analysis. Following this preprocessing step, the Seurat package (version 2.3) was employed for both dimensionality reduction and cluster identification. The normalization and scaling of gene expression data were accomplished using the NormalizeData and ScaleData functions, respectively. Subsequently, the FindVariableFeatures function facilitated the selection of the 2000 genes demonstrating the most significant variation for principal component analysis. Cluster identification was achieved through the FindClusters function, which partitioned the genes into distinct groups. To mitigate batch effects across samples, the Harmony package was utilized. The refined dataset was then visualized in a two-dimensional space employing Uniform Manifold Approximation and Projection techniques, thereby providing insightful representations of the underlying cellular heterogeneity.

### Chromosomal copy-number variations estimation

The estimation of chromosomal copy-number variations (CNVs) was conducted utilizing the 'inferCNV' R package. A diverse reference panel comprised of B cells, T cells, NK cells, macrophages, and other immune cells facilitated this analysis. Quantification of CNV scores for each subcluster was achieved by aggregating the CNV levels observed across the constituent cells.

### Quantification and statistical analysis

Statistical comparisons were performed using unpaired two-tailed Student's $t$-tests, Mann–Whitney test, one-way ANOVA or two-way ANOVA. Statistical analyses were performed using GraphPad Prism 9.0 software (GraphPad Software, La Jolla, CA). Data are presented as mean ±the standard error of the mean (SEM). Statistical significance was defined as $p < 0.05$.

## Acknowledgements

This study was supported by grants from the Ministry of Science and Technology of China 2022YFA1104003 and 2021YFC2501000; National Natural Science Foundation of China grants 82172972, 82230094, 81972616, 81972609, and 81772840; Peking University Medicine Sailing Program for Young Scholars' Scientific & Technological Innovation BMU2024YFJHPY004; the Fundamental Research Funds for the Central Universities; Postdoctoral Fellowship Program of CPSF GZC20230159. We would like to thank TissueGnostics Asia Pacific Limited for their technical support on multiple immunohistochemistry staining.

## Additional information

### Funding

| Funder | Grant reference number | Author |
|---|---|---|
| Ministry of Science and Technology of the People's Republic of China | 2022YFA1104003 | Hongquan Zhang |
| Ministry of Science and Technology of the People's Republic of China | 2021YFC2501000 | Hongquan Zhang |
| National Natural Science Foundation of China | 82172972 | Jun Zhan |

| Funder | Grant reference number | Author |
| --- | --- | --- |
| National Natural Science Foundation of China | 82230094 | Hongquan Zhang |
| National Natural Science Foundation of China | 81972616 | Hongquan Zhang |
| National Natural Science Foundation of China | 81972609 | Jun Zhan |
| National Natural Science Foundation of China | 81772840 | Xiaojing Guo |
| Peking University Health Science Center | BMU2024YFJHPY004 | Miao Yu |
| Postdoctoral Fellowship Program of CPSF | GZC20230159 | Miao Yu |

The funders had no role in study design, data collection and interpretation, or the decision to submit the work for publication.

## Author contributions

Weijie Wu, Data curation, Formal analysis, Validation, Investigation, Visualization, Writing – review and editing; Miao Yu, Data curation, Formal analysis, Funding acquisition, Validation, Investigation, Writing - original draft; Qianchen Li, Lei Zhang, Investigation, Visualization; Yiqian Zhao, Yi Sun, Zhenbin Wang, Yuqing Gong, Chenying Liu, Jing Zhang, Yan Tang, Investigation; Wenjing Wang, Funding acquisition, Investigation; Xiaojie Xu, Xiaojing Guo, Funding acquisition, Writing – review and editing; Jun Zhan, Funding acquisition, Project administration, Writing – review and editing; Hongquan Zhang, Conceptualization, Funding acquisition, Project administration, Writing – review and editing

## Author ORCIDs

Weijie Wu http://orcid.org/0009-0001-4621-7243
Miao Yu http://orcid.org/0000-0002-2256-0231
Hongquan Zhang https://orcid.org/0000-0001-8193-0899

## Ethics

Mice were housed in a pathogen-free animal facility at Laboratory Animal Center of Peking University Health Science Center with a 12 h light/dark cycle, constant temperature and humidity, and fed standard rodent chow and water ad libitum. All animal experiments were approved by the Peking University Biomedical Ethics Committee and the approval number is BCJB0104.

Reviewer #1 (Public review): https://doi.org/10.7554/eLife.101888.3.sa1
Reviewer #2 (Public review): https://doi.org/10.7554/eLife.101888.3.sa2
Author response https://doi.org/10.7554/eLife.101888.3.sa3

# Additional files

## Supplementary files

Supplementary file 1. Mass spectrometry (MS) analyses of HEK293A cells transiently expressing Flag-FRMD8.

Supplementary file 2. Primers for genotyping, RNA silencing, and qRT-PCR.

Supplementary file 3. Patient information of tissue microarray.

MDAR checklist

## Data availability

The accession number for the scRNA-seq data reported in this paper is Gene Expression Omnibus (GEO): GSE244582.

The following dataset was generated:

| Author(s) | Year | Dataset title | Dataset URL | Database and Identifier |
|---|---|---|---|---|
| Wu W, Yu M, Zhan J, Zhang H | 2025 | Loss function of tumor suppressor FRMD8 confers resistance to tamoxifen therapy via a dual mechanism | https://www.ncbi.nlm.nih.gov/geo/query/acc.cgi?acc=GSE244582 | NCBI Gene Expression Omnibus, GSE244582 |

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
