## [Editor Report · eLife Assessment]

The research presents **valuable** findings on the impact of FRMD8 loss on tumor progression and resistance to tamoxifen therapy. Through a series of **convincing** and systematic experiments, the author thoroughly investigates the role of FRMD8 in breast cancer and its underlying regulatory mechanisms. The study confirms that FRMD8 holds potential as a therapeutic target for reversing tamoxifen resistance, offering helpful insights for future treatment strategies.

---

## [Referee Report · Reviewer #1 (Public review)]

Summary:

Tamoxifen resistance is a common problem in partially ER-positive patients undergoing endocrine therapy, and this manuscript has important research significance as it is based on clinical practical issues. The manuscript discovered that the absence of FRMD8 in breast epithelial cells can promote the progression of breast cancer, thus proposing the hypothesis that FRMD8 affects tamoxifen resistance and validated this hypothesis through a series of experiments. The manuscript has certain theoretical reference value.

Strengths:

At present, research on the role of FRMD8 in breast cancer is very limited. This manuscript leverages the *MMTV-Cre^+^*; *Frmd8^fl/fl^*; *PyMT* mouse model to study the role of FRMD8 in tamoxifen resistance, and single-cell sequencing technology discovered the interaction between FRMD8 and ESR1. At the mechanistic level, this manuscript has demonstrated two ways in which FRMD8 affects ERα, providing some new insights into the development of ER-positive breast cancer in patients who are resistant to tamoxifen.

Limitations:

Whether FRMD8 can become a biomarker should be verified in large clinical samples or clinical data.

---

## [Referee Report · Reviewer #2 (Public review)]

Summary:

The manuscript presents a valuable finding on the impact of FRMD8 loss on tumor progression and the resistance to tamoxifen therapy. The author conducted systematic experiments to explore the role of FRMD8 in breast cancer and its potential regulatory mechanisms, confirming that FRMD8 could serve as a potential target to revere tamoxifen resistance.

The research is logically coherent and persuasive. The results support their conclusions and have achieved the research objectives.

---

## [Author Response]

The following is the authors’ response to the original reviews.

**Public Reviews:**

**Reviewer #1 (Public review):**
Summary:Tamoxifen resistance is a common problem in partially ER-positive patients undergoing endocrine therapy, and this manuscript has important research significance as it is based on clinical practical issues. The manuscript discovered that the absence of FRMD8 in breast epithelial cells can promote the progression of breast cancer, thus proposing the hypothesis that FRMD8 affects tamoxifen resistance and validating this hypothesis through a series of experiments. The manuscript has a certain theoretical reference value.Strengths:At present, research on the role of FRMD8 in breast cancer is very limited. This manuscript leverages the *MMTV-Cre^+^*; *Frmd8^fl/fl^*; *PyMT* mouse model to study the role of FRMD8 in tamoxifen resistance, and single-cell sequencing technology discovered the interaction between FRMD8 and ESR1. At the mechanistic level, this manuscript has demonstrated two ways in which FRMD8 affects ERα, providing some new insights into the development of ER-positive breast cancer in patients who are resistant to tamoxifen.Weaknesses:This manuscript repeatedly emphasizes the role of FRMD8/FOXO3A in tamoxifen resistance in ER-positive breast cancer, but the specific mechanisms have not yet been fully elucidated. Whether FRMD8 can become a biomarker should be verified in large clinical samples or clinical data.

We appreciate your recognition and valuable suggestions. The proliferation of ERα-positive breast cancer cells is contingent upon the expression of ERα. Tamoxifen, a selective estrogen receptor modulator, competitively binds to ERα, thereby inhibiting the activation of the proliferation signaling pathway. Previous studies have demonstrated that the downregulation of ERα expression results in a reduction in the sensitivity of breast cancer cells to tamoxifen (PMID: 15894097; PMID: 922747). Our study revealed the molecular mechanism by which FRMD8 regulates ERα expression through FOXO3A and UBE3A, and thus FRMD8 deficiency is a cause of tamoxifen treatment resistance.

In this study, our results showed that low expression of FRMD8 predicts poor prognosis in breast cancer patients. We agree with this reviewer and will validate the role of FRMD8 in more patient samples and expand its application in different cancer types.

**Reviewer #2 (Public review):**
Summary:The manuscript presents a valuable finding on the impact of FRMD8 loss on tumor progression and the resistance to tamoxifen therapy. The author conducted systematic experiments to explore the role of FRMD8 in breast cancer and its potential regulatory mechanisms, confirming that FRMD8 could serve as a potential target to revere tamoxifen resistance.Strengths:The majority of the research is logically clear, smooth, and persuasive.Weaknesses:Some research in the article lacks depth and some sentences are poorly organized.

Thank you for your helpful suggestion. We have carefully revised the manuscript again.

**Recommendations for the authors:**

**Reviewer #1 (Recommendations for the authors):**
This manuscript suggests that the resistance of tamoxifen in breast cancer is linked to the loss of function of FRMD8. This is a relatively good and valuable contribution. However, there are several points that confused me.(1) The subfigures with important conclusions should include quantitative analysis, for example, Figure 4D, 4E, and 6A. In Figure 6F, which subtypes of normal and tumor tissues were investigated.

Thank you for your helpful suggestions. We have quantified the bands in Figure 4D, 4E, and 6A and labelled them in the figures.

We have also provided details of the tumor samples in Table S3 and the “Materials and Methods” section. The majority of tumor tissues are invasive ductal carcinomas.

(2) In the luminal epithelium-specific Frmd8 knockout mice (*MMTV-Cre^+^*; *Frmd8^fl/fl^*), the authors demonstrated that the loss of FRMD8 promotes the growth of breast tumors. In Figure 3A, the expression of ERα and PR in tumors is nearly negative. However, why was the validation of the mechanism performed in breast tumor cell lines and not in epithelial cells?

Thanks for the question. Early-stage mammary tumors in *MMTV-PyMT* mice express ERα, while ERα is negative in advanced tumors of *MMTV-PyMT* mice. Figure 3A shows the results of tumors from four-month-old mice. Meanwhile, our supplementary results showed that loss of Frmd8 decreased ERα expression also in normal and atypical hyperplasia mammary tissues from 7-week-old *MMTV-PyMT* mice, when the mice had no palpable tumors and ERα is positive (Fig. S3E). We believe that the absence of FRMD8 contributes to the acceleration of the malignant progression during the dynamic evolution of breast cancer. Limited by the difficulty of transfection in breast normal epithelial cell line (MCF10A), we explored the subsequent mechanisms mainly in breast cancer cells and HEK293, a human embryonic kidney cell line. Besides, Figure S3E also showed the regulation of ERα expression by Frmd8 in mouse mammary

epithelial cells.

(3) To explore the mechanism by which FRMD8 inhibits ERα degradation, what is the reason for choosing HEK293A?

Thank you for the good question. HEK293 cell line is commonly used in mechanistic studies. We also employed the breast cancer cell line T47D to verify the observations in HEK293 cells. Furthermore, the mass spectrometry result of HEK293A cells presented in Figure 5E was an additional experiment performed when we were exploring the regulation of the cell cycle by FRMD8, which is published in Cell Reports (PMID: 37527040). Based on the mass spectrometry result, we assumed that FRMD8 may influence ERα degradation mediated by UBE3A.

**Reviewer #2 (Recommendations for the authors):**
Introduction(1) In order for the reader to better understand the content of the article, it is better to briefly describe the role of ERα in the progression of breast cancer.

Thank you for your suggestion. We have provided a brief description of the role of ERα in the introduction of revised manuscript:

“ERα is a ligand-activated transcription factor that is activated by oestrogen, and promotes cell proliferation during breast cancer development (Harbeck et al., 2019).”

(2) As ESR1 is mentioned in the second paragraph, a brief description of the relationship between ESR1 and ERα can make the article more logical.

Thank you for the suggestion. We have added the description in the introduction:

“Multiple transcription factors, such as AP-2γ, FOXO3, FOXM1, and GATA3, have been reported to bind to the promoter region of *ESR1*, the gene encoding ERα, and participate in transcriptional regulation of *ESR1*(Jia et al., 2019; Koš et al., 2001).”

(3) In the text, there are two variations of the term FRMD8: 'FRMD8' and 'Frmd8'. It is best to standardize on one form throughout the document.

We apologize for any confusion. The terms "FRMD8" and "Frmd8" are used to indicate proteins derived from human and mouse, respectively.

Results(4) In Figure 2L, there is no noticeable difference in the expression levels of Pgr and Esr1 between the Cre+ tumor and Cre- tumor groups. Figure S2E is more suitable for inclusion in the main text compared to Figure 2L.

Thank you for this suggestion. ERα and PR are positive in early-stage mammary tumors of *MMTV-PyMT* mice, while ERα and PR are gradually lost as the tumor progresses. In figure 2, mammary tumors from 4-month-old *MMTV-PyMT* mice were subjected to scRNA-seq analysis. Since the expression of ERα was very low in tumor cells at this time, there appears to be no difference between the two groups. We have exchanged Figure 2L and Figure S2E in the manuscript.

(5) The CNV score can be used to assess the malignancy of cells, it would be better to compare the malignancy levels between the two groups.

This is a very good suggestion. However, copy number variations usually occur randomly and have a high degree of heterogeneity. Due to the limited sample size in our study, we did not compare the difference between the two groups.

(6) Enrichment analysis is crucial for single-cell sequencing studies. It is recommended to perform differential gene analysis and enrichment analysis between the Cre+ and Cre- groups to further explore the impact of FRMD8 deficiency on the functions of malignant cells.

Thank you for your suggestion. We have performed differential gene analysis and biological process enrichment analysis on the results of scRNA sequence using the gene ontology (GO) database. Our results showed that upregulated genes in luminal progenitor (Lp) epithelial cells were enriched in epithelial cell proliferation and transmembrane receptor protein serine/threonine kinase signaling pathways, suggesting that Frmd8 deficiency significantly promotes epithelial cells proliferation in *MMTV-PyMT* mice.

(7) The coherent logic in lines 300 to 308 should be that FRMD8 is expressed at higher levels in normal Hsd epithelial cells in mice, hence further verification was conducted to examine the expression levels of FRMD8 in various human breast cancer cell lines.

We have revised the figures and text as suggested.

Discussion(8) In lines 352 to 360, the background narrative in the first half seems to have little connection with the research findings in the second half; it is suggested to reorganize the language of this section.

Thank you for the advice. We have rewritten this paragraph in the manuscript:

“In *MMTV-PyMT* mice, early-stage mammary tumors express ERα and PR, but these receptors are gradually lost as the tumor progresses (Lapidus et al., 1998). Our scRNA-seq results revealed that mammary tumor epithelial cells in *MMTV-PyMT* mice fall into four clusters, with only Hsd epithelial cells showing ERα and PR expression. Additionally, Hsd epithelial cells exhibited the lowest CNV score, indicating a closer resemblance to normal epithelial cells. The loss of Frmd8 reduced the proportion of Hsd epithelial cells and led to a downregulation of ERα and PR expression, implying that Frmd8 deficiency promotes the loss of luminal features in the mammary gland and accelerates mammary tumor progression.”

(9) As stated in the result section, the depletion of FRMD8 may lead to the decrease of the Hsd epithelial cells proportion, it might be beneficial to discuss the significance of this finding.

We have added a discussion of the Hsd epithelial cell proportion in the third paragraph of this section (please refer to the above question (8)).

Figures(10) The structural layout of Figure 4 should be reorganized to make it more aesthetically pleasing.

Thank you for this suggestion. We have rearranged Figure 4 as suggested.